# A Systems Hypothesis of Lipopolysaccharide-Induced Vitamin Transport Suppression and Metabolic Reprogramming in Autism Spectrum Disorders: An Open Call for Validation and Therapeutic Translation

**DOI:** 10.3390/metabo15060399

**Published:** 2025-06-13

**Authors:** Albion Dervishi

**Affiliations:** Anaesthesiology and Intensive Care Medicine, medius Clinic Ostfildern-Ruit-Academic Teaching Hospital of the University of Tübingen, Hedelfinger Str. 166, 73760 Ostfildern, Germany; albiondervishi@gmail.com

**Keywords:** autism spectrum disorder (ASD), systems biology, mitochondrial dysfunction, SLC5A6, SLC19A2, biotin, thiamine, pantothenic acid, transcriptomics, lipopolysaccharide (LPS), metabolic reprogramming, redox adaptation, precision medicine, multivitamin transporters, neurodevelopment

## Abstract

**Background:** Autism spectrum disorder (ASD) is increasingly linked to systemic metabolic dysfunction, potentially influenced by gut–brain axis dysregulation, but the underlying mechanisms remain unclear. **Methods:** We developed Personalized Metabolic Margin Mapping (PM^3^), a computational systems biology framework, to analyze RNA-seq data from 12 ASD and 12 control postmortem brain samples. The model focused on 158 curated metabolic genes selected for their roles in redox balance, mitochondrial function, neurodevelopment, and gut–brain interactions. **Results:** Using unsupervised machine learning (Isolation Forest) to detect outlier expression patterns, Euclidean distance, and percent expression difference metrics, PM^3^ revealed a consistent downregulation of glycolysis (e.g., −5.4% in PFKM) and mitochondrial enzymes (e.g., −12% in SUCLA2). By incorporating cofactor dependency and subcellular localization, PM^3^ identified a coordinated suppression of multivitamin transporters (e.g., −4.5% in SLC5A6, −3.5% in SLC19A2), potentially limiting cofactor availability and compounding energy deficits in ASD brains. **Conclusions:** These findings suggest a convergent metabolic dysregulation signature in ASD; wherein the subtle suppression of cofactor-dependent pathways may impair energy metabolism and neurodevelopment. We propose that chronic microbial lipopolysaccharide (LPS) exposure in ASD suppresses vitamin transporter function, initiating mitochondrial dysfunction and transcriptomic reprogramming. Validation in LPS-exposed systems using integrated transcriptomic–metabolomic analysis is warranted.

## 1. Introduction

Autism spectrum disorder (ASD) is a neurodevelopmental condition defined by a wide range of behavioral symptoms. While extensive research has identified many gene variants associated with ASD, mechanisms connecting genotype to phenotype are still unknown [1,2]. In addition to neural circuit alterations, increasing evidence highlights systemic metabolic abnormalities as key contributors to ASD pathophysiology. These include mitochondrial dysfunction, oxidative stress, amino acid imbalances, and impaired detoxification processes [3,4].

Transcriptomic and metabolomic studies of ASD brain tissue have consistently shown reduced oxidative phosphorylation, glutathione depletion, and the repression of mitochondrial gene expression [4,5,6]. Together, these findings suggest a broader pattern of bioenergetic collapse and redox imbalance. Notably, such metabolic alterations may arise not only from genetic predisposition but also from environmental factors—particularly microbial metabolites and immune activation—that interact with host metabolism during critical periods of brain development [7,8,9].

A notable environmental trigger is lipopolysaccharide (LPS), a pro-inflammatory endotoxin derived from Gram-negative gut bacteria. LPS is frequently elevated in individuals with autism and gut dysbiosis [10] and is known to impair nutrient transporters, mitochondrial enzymes, and blood–brain barrier integrity, potentially limiting cerebral access to essential cofactors such as biotin, pantothenic acid, thiamine, and lipoic acid [11,12,13].

To investigate this hypothesis, we developed a systems biology model called Personalized Metabolic Margin Mapping (PM^3^). This model focuses on cofactor-dependent genes involved in energy production, redox balance, amino acid metabolism, and methylation. PM^3^ integrates subcellular localization (cytosolic vs. mitochondrial), cofactor dependencies, and pathway context to identify transcriptional signatures of metabolic stress. In this study, we apply PM^3^ to RNA-seq data from prefrontal cortex samples of individuals with ASD and matched controls to assess redox-adaptive reprogramming and mitochondrial vulnerability.

## 2. Methods

### 2.1. Transcriptomic Dataset Acquisition

We utilized publicly available, normalized RNA-seq expression data from the study by Parikshak et al. (GSE64018), consisting of postmortem cortical tissue from individuals diagnosed with autism spectrum disorder (ASD, *n* = 12) and neurotypical controls (*n* = 12) [14]. Samples were obtained from the superior temporal gyrus (Brodmann areas 41/42/22), with gray matter retained across all cortical layers. RNA extraction, rRNA depletion, and size selection were performed as described in the original study, ensuring high-integrity mRNA for transcriptomic analysis.

### 2.2. PM^3^ Gene Model Construction

We developed a curated metabolic gene model, Personalized Metabolic Margin Mapping (PM^3^), comprising 158 genes selected for their involvement in essential biochemical pathways, namely energy production, redox regulation, amino acid metabolism, methylation, and detoxification. Each gene was annotated with its corresponding enzyme, metabolic domain, cofactor dependency, and subcellular localization (cytoplasmic or mitochondrial). Subcellular compartmentalization was determined based on UniProt annotations (https://www.uniprot.org, accessed on 11 May 2025) and supplemented by a literature review.

PM^3^ is a systems biology framework designed to identify coordinated transcriptional shifts and compartment-specific metabolic vulnerabilities in ASD. Unlike conventional differential gene expression approaches emphasizing statistical significance at the single-gene level, PM^3^ captures subtle systems-level disruptions across biochemically linked pathways. This model detects nonlinear transcriptomic patterns that may be missed by standard analytical approaches by integrating information on cofactor usage, subcellular localization, and the context of metabolic domains.

Unlike conventional differential expression tools such as DESeq2 or edgeR, which prioritize statistical significance at the individual gene level, the PM^3^ framework is designed to detect hidden coordination across metabolically linked genes. It reveals subtle but biologically coherent transcriptional patterns that may be overlooked by *p*-value-driven analyses.

### 2.3. Differential Expression and Divergence Metrics

To evaluate gene-specific expression changes between ASD and control groups, we computed the following: 1. Percent Expression Difference (%Δ): Calculated from group means to indicate directional upregulation or downregulation. 2. Mean Euclidean Distance (ED): Computed across all samples per gene to quantify variability and divergence between ASD and control expression distributions. 3. Area Under the ROC Curve (AUC): Derived from unsupervised Isolation Forest models to quantify each gene’s ability to distinguish ASD from control states, serving as a transcriptomic anomaly score.

No threshold was applied for log2 fold change, as our focus was to capture subtle but coordinated metabolic shifts rather than statistically maximal differentials. Due to the limited sample size and the unsupervised nature of the analysis, formal statistical tests were not applied.

Both percentage difference (%Δ) and log_2_ fold change (log_2_FC) were computed for all genes to reflect the magnitude and direction of transcriptional shifts. While %Δ was used for visual clarity in figures, log_2_FC values are included in Table 1 and Appendix A for alignment with standard transcriptomic practices.

### 2.4. Machine Learning and Gene-Level Anomaly Detection

An unsupervised Isolation Forest algorithm was implemented using the H2O.ai machine learning platform (v3.42.0.2) to compute anomaly scores for each gene across all samples. These scores captured nonlinear expression deviations and potential transcriptional outliers contributing to metabolic instability. Additionally, K-means clustering was applied to the anomaly score matrix to identify distinct gene clusters reflecting metabolic expression patterns. AUC values were then calculated to assess the discriminatory strength of each gene in separating ASD from control states.

### 2.5. Software and Visualization Tools

All analyses were performed using R (v4.3.0). Key packages included tidyverse [15], ggplot2 [16], ComplexHeatmap [17], and H2O [18]. Full gene annotations—including metabolic domain, cofactor dependency, and expression metrics—are provided in Appendix A

## 3. Results

### 3.1. Divergence of Metabolic Gene Expression in ASD Cortex

To test our hypothesis that the lipopolysaccharide (LPS)-mediated suppression of vitamin transporters contributes to metabolic dysfunction in autism, we applied the PM^3^ model to transcriptomic data from postmortem prefrontal cortex samples (n = 12 ASD, n = 12 controls). Using a panel of 158 curated metabolic genes, we quantified transcriptional divergence through unsupervised anomaly detection with Isolation Forest, supported by percent expression difference (%Δ), Euclidean distance, and AUC scores.

Out of the 158 genes, 20 showed AUC values above 0.70, indicating robust discriminatory power between ASD and control samples (Table 1). Among the highest-ranking genes were SUCLA2 (AUC = 0.88, %Δ = −11.93), GOT1 (0.86, −11.82), SLC25A11 (0.85, −7.99), and CKMT1A (0.84, −16.9), reflecting substantial mitochondrial and redox axis alterations.

In contrast, certain genes exhibited moderate upregulation. Notably, GPT2 (AUC = 0.83, %Δ = +4.64), SHMT1 (AUC = 0.77, %Δ = +7.55), and SUCLG2 (AUC = 0.69, %Δ = +13.83) showed positive expression shifts. Full expression metrics, including AUC, %Δ, metabolic domain, and cofactor dependency, are provided in Appendix A.

### 3.2. Systems-Level Visualization of Metabolic Dysregulation in ASD

To enhance clarity, we visualized the directional dysregulation of these genes using alluvial diagrams (Figure 1A,B). These plots map genes across three layers, namely the metabolic domain, pathway, and gene/enzyme name. Figure 1A illustrates upregulated genes associated with compensatory functions, such as glycolysis and one-carbon metabolism. Figure 1B highlights downregulated genes involved in mitochondrial function, redox balance, and cofactor transport.

Figure 2 presents a systems-level schematic summarizing transcriptional shifts across key biochemical pathways, including mitochondrial metabolism, redox cycling, and detoxification. Genes are color-coded by expression direction, highlighting a pattern of energy suppression and compensatory redox adaptation.

### 3.3. Coordinated Downregulation of Cofactor Transporters and Related Enzymes

To assess the impact of microbial signaling on vitamin transporter expression, we examined genes encoding cofactor transporters in the PM^3^ model. Figure 3 illustrates % expression change and inter-sample variability (Euclidean distance) across cytosolic and mitochondrial compartments.

Key transporters exhibited mild but coordinated transcriptional downregulation in the ASD cortex. SLC5A6, encoding the sodium-dependent multivitamin transporter (SMVT), was reduced by 4.5%, while SLC19A2, responsible for thiamine pyrophosphate (TPP) uptake, was downregulated by 3.5%. In contrast, SLC19A3, a thiamine transporter variant, was modestly upregulated (+3.9%).

Differential expression was also observed in pantothenate kinase isoforms: PANK1 was downregulated by 11.8%, whereas PANK2 was slightly upregulated (+1.7%), suggesting a possible compartment-specific adaptation in CoA biosynthesis.

These shifts affected enzymes dependent on the corresponding cofactors, including the pyruvate dehydrogenase complex (PDHA1, DLAT, DLD), α-ketoglutarate dehydrogenase (OGDH), and branched-chain keto acid dehydrogenase (BCKDH), which rely on thiamine, lipoic acid, CoA, and biotin.

To aid interpretation, Appendix A summarizes the function, cofactor specificity, and subcellular localization of the cofactor transporters analyzed in this study.

### 3.4. Domain-Specific Metabolic Disruption in ASD Cortex

To assess functional grouping patterns within the PM^3^ gene set, we stratified the 158 curated metabolic genes into three major domains as follows:(A).Glycolysis and pentose phosphate pathway (PPP);(B).Redox, sulfur, and one-carbon metabolism;(C).Mitochondrial metabolism.

Figure 4 visualizes the percent expression difference (%Δ) and the mean Euclidean distance for each gene, revealing domain-specific transcriptional shifts and inter-sample variability between ASD and control cortices.

In the glycolysis/PPP domain, PFKP (−5.4%) and PKM2 (−5.1%) were moderately downregulated. In contrast, two PPP-associated genes, PMM2 (+30.3%) and SORD (+24.5%), were markedly upregulated, suggesting partial activation of non-glycolytic glucose metabolism (Figure 4A).

The redox/sulfur/one-carbon domain showed divergent expression patterns. Sulfur detoxification genes ETHE1 (+23%) and TST (+0.4%) were upregulated, whereas SUOX (−6.6%) was downregulated. One-carbon enzymes such as SHMT2 (+12%) and SHMT1 (+7.55%) showed increased expression, while UGP2, required for glucuronidation, was downregulated (−5.6%) (Figure 4B).

In the mitochondrial domain, transcriptional suppression was observed across core enzyme complexes. PDHA1 (−2.25%), DLAT, DLD, IDH3A (−8.33%), IDH3B (−8.13%), and OGDH (−3.63%) were among the most affected. Additionally, propionate detoxification enzymes PCCA (−5.5%), PCCB (−6.8%), and MUT (−7.9%) were also suppressed (Figure 4C).

Together, these data highlight selective domain-specific vulnerability in the ASD cortex, most notably in mitochondrial energy metabolism, glucuronidation, and glycolytic flux.

## 4. Discussion

The PM^3^ model revealed coordinated transcriptional shifts across metabolic pathways, indicating a systems-wide reorganization of cortical metabolism in autism. Upregulated antioxidant (e.g., NQO1) and glycosylation genes (e.g., PMM2), alongside downregulated mitochondrial energy enzymes (e.g., CKMT1A) and folate transporters (e.g., SLC25A32), point to disrupted glutamate metabolism, mitochondrial transport, and energy buffering—consistent with prior reports of mitochondrial dysfunction [3,6]. Unlike conventional differential expression methods, PM^3^ detects subtle yet coordinated shifts by integrating cofactor dependencies and subcellular localization. It captures patterns—such as the suppression of SLC5A6 and SLC19A2—that are often missed by single-gene approaches.

These metabolic alterations may converge with immune signaling pathways. Recent studies have identified a subset of ASD cases with upregulated LPS-responsive genes (e.g., MAPK8, TNFSF12), implicating microbial endotoxin exposure as a potential upstream trigger of metabolic and immune reprogramming in the autistic brain [10].

### 4.1. Suppression of Cofactor Transporters in ASD Cortex

The observed transcriptional suppression of cofactor transporters such as SLC5A6 and SLC19A2 supports a mechanistic model in which microbial lipopolysaccharide (LPS) exposure contributes to acquired nutrient deficiencies in the ASD brain. Previous studies have demonstrated that LPS can downregulate SLC5A6 expression in colonic epithelial cells via casein kinase 2 (CK2)-mediated signaling pathways [11]. We propose that similar LPS-sensitive regulatory mechanisms may operate within the central nervous system, leading to localized reductions in biotin, pantothenic acid, and lipoic acid availability despite adequate systemic levels.

Thiamine transport may be particularly vulnerable, as SLC19A2 is known to be inhibited through two distinct LPS-driven mechanisms, namely TLR4/NF-κB-dependent transcriptional repression [12] and the protein kinase A (PKA)-mediated disruption of membrane transporter expression [13]. These complementary pathways could amplify intracellular thiamine pyrophosphate (TPP) deficiency, especially in metabolically demanding tissues such as the brain.

Interestingly, the slight upregulation of SLC19A3 (+3.9%) may reflect a compensatory stress response, consistent with prior findings in mitochondrial disease models where SLC19A3 expression is induced under oxidative stress [19].

The differing regulation of PANK1 and PANK2 highlights a complex adaptation that may help maintain mitochondrial CoA synthesis despite limited pantothenate availability in the cytosol. These changes might act as a transcriptional response to interrupted vitamin delivery, but they could also result in imbalanced CoA levels across different cellular compartments.

### 4.2. Glycolytic Downregulation and Citrate–Pyruvate Feedback

The PM^3^ model revealed suppressed glycolytic flux in the autism cortex, with a reduced expression of key enzymes such as PFKP and PKM2. Figure 4A illustrates these domain-specific gene expression patterns, showing percent expression differences and Euclidean distances for glycolysis and pentose phosphate pathway genes, color-coded by metabolic domain. This suppression may result from elevated intracellular pyruvate and citrate levels, which allosterically inhibit PFK and PKM2. The upregulation of SLC25A1 from 15% and downregulation of ACLY and ACACB suggest cytosolic citrate accumulation, imposing a glycolytic brake. Concurrently, the increased expression of GPT2 and GLUD1/2 indicates a compensatory transamination pathway (ALT–GLUD loop) to manage pyruvate overflow, recycling glutamate and α-ketoglutarate while minimizing lactate production, akin to mechanisms observed in cancer metabolism [20]. This metabolic rerouting highlights a pre-Warburg adaptation in autism, potentially driven by upstream mitochondrial stress and cofactor deficiencies identified by the PM^3^ model.

### 4.3. Mitochondrial Bottlenecks in the TCA Cycle

Key enzymes of the TCA cycle were transcriptionally suppressed in the ASD cortex, including the pyruvate dehydrogenase complex (PDHA1, DLAT, DLD), isocitrate dehydrogenases (IDH3A, IDH3B, IDH2), and the α-ketoglutarate dehydrogenase complex (OGDH, DLST, DLD). Average downregulation was ~5.6% for PDH, ~8.2% for IDHs, and ~6.3% for OGDH. These consistent reductions reflect a coherent energetic downshift that constrains mitochondrial throughput and redox cycling.

These molecular signatures are consistent with the biochemical evidence of suppressed mitochondrial respiration in ASD brains [21,22,23]. Collectively, these observations reinforce the hypothesis of a bioenergetic bottleneck driven by both cofactor deficiency and transcriptional inhibition of mitochondrial enzyme complexes (Figure 4C).

### 4.4. Propionic Acid as a Mitochondrial Toxin

The PM^3^ model identified reduced transcription of biotin- and B12-dependent enzymes involved in propionic acid (PPA) detoxification, including PCCA, PCCB, and MUT (Figure 2 and Figure 4C). This impairment suggests a bottleneck in propionyl–CoA clearance, exacerbating mitochondrial stress in autism. These findings align with evidence that PPA, a gut-derived short-chain fatty acid, disrupts mitochondrial function [24,25]. The PM^3^ model’s systems-level approach reveals how cofactor insufficiency and transcriptional repression converge to limit detoxification capacity, supporting the hypothesis that microbial metabolites contribute to metabolic dysregulation in the disorder. Further studies are needed to quantify PPA levels in autism and validate these transcriptional patterns.

### 4.5. Upregulation of Sulfur Metabolism Pathways in ASD

In autism spectrum disorder (ASD), gut dysbiosis and xenobiotic burden place increasing stress on detoxification pathways. Suppressed glycolysis may compromise glucuronidation, as reflected in reduced UGP2 expression, limiting UDP–glucuronic acid synthesis. Concurrently, the overgrowth of β-glucuronidase-producing bacteria promotes deconjugation and toxin reabsorption [26,27,28].

This metabolic strain shifts the detoxification load toward sulfation, which depends on intracellular sulfate and ATP-driven PAPS synthesis. The upregulation of ETHE1 and TST suggests increased H_2_S clearance, consistent with elevated urinary sulfur metabolites in ASD [29,30].

However, the downregulation of SUOX—essential for sulfite oxidation—may hinder clearance and elevate oxidative stress [31,32]. Together, these changes reflect a disrupted balance between microbial metabolism, sulfur detoxification, and host energy regulation (Figure 4B).

### 4.6. One-Carbon Metabolism Disruption in ASD

The brain’s one-carbon metabolism depends on serine biosynthesis from glycolytic intermediates [33,34]. In the ASD cortex, a reduced expression of PHGDH—the rate-limiting enzyme initiating serine synthesis—likely reflects diminished glycolytic input, potentially constraining one-carbon flux.

Despite this upstream limitation, downstream enzymes such as SHMT1, SHMT2, and PSPH appear upregulated, suggesting a compensatory response to sustain methyl group transfer. Concurrently, increased glycine degradation via GLDC and GCSH, with a relative decrease in AMT, indicates an alternative strategy to maintain folate-mediated one-carbon units under serine-limited conditions.

These adaptations point to a secondary methylation bottleneck—not from direct defects in one-carbon enzymes but from upstream metabolic stress. The resulting serine–glycine imbalance may impair epigenetic programming and contribute to neurodevelopmental dysregulation in ASD.

### 4.7. Therapeutic Implications

If validated, this model highlights SLC5A6 and SLC19A2 as potential targets for restoring cofactor delivery in ASD. While specific interventions remain to be determined, this framework opens new possibilities for translational research—ranging from nutrient-based strategies to microbiome-directed approaches. The identification of vitamin transporter suppression as a convergent mechanism invites further exploration by both academic and industry partners to develop targeted solutions for metabolic restoration in ASD.

### 4.8. Study Limitations and Future Directions

This study is based on retrospective transcriptomic data, and a direct correlation between plasma lipopolysaccharide (LPS) levels and tissue-specific expression of SLC5A6, SLC19A2, or other PM^3^ model genes could not be established. In particular, brain and peripheral tissues may respond differently to microbial signals, and such spatial heterogeneity cannot be resolved through bulk RNA-seq alone. Moreover, cofactor levels (e.g., biotin, pantothenic acid, thiamine) and post-transcriptional regulation were not evaluated, potentially underestimating functional deficits. These limitations highlight the need for prospective studies incorporating multi-omics profiling, LPS challenge models, and metabolite quantification across relevant tissues to validate and extend the metabolic cascade proposed here.

## 5. Conclusions

This study presents a systems-level view of metabolic dysfunction in ASD, proposing that microbial lipopolysaccharide (LPS) inhibits key cofactor transporters (SLC5A6, SLC19A2), leading to a downstream suppression of glycolysis, mitochondrial function, and one-carbon metabolism. We show that this inhibition contributes to reduced transcription genes for glucuronidation and serine synthesis, shifting detoxification toward energy-intensive sulfation and mitochondrial sulfur pathways. These adaptations reveal a coordinated metabolic response with implications for neurodevelopment and epigenetic dysregulation. These findings warrant a clinical investigation of nutrient transport and mitochondrial function in biomarker-stratified ASD subtypes.

## Figures and Tables

**Figure 1 metabolites-15-00399-f001:**
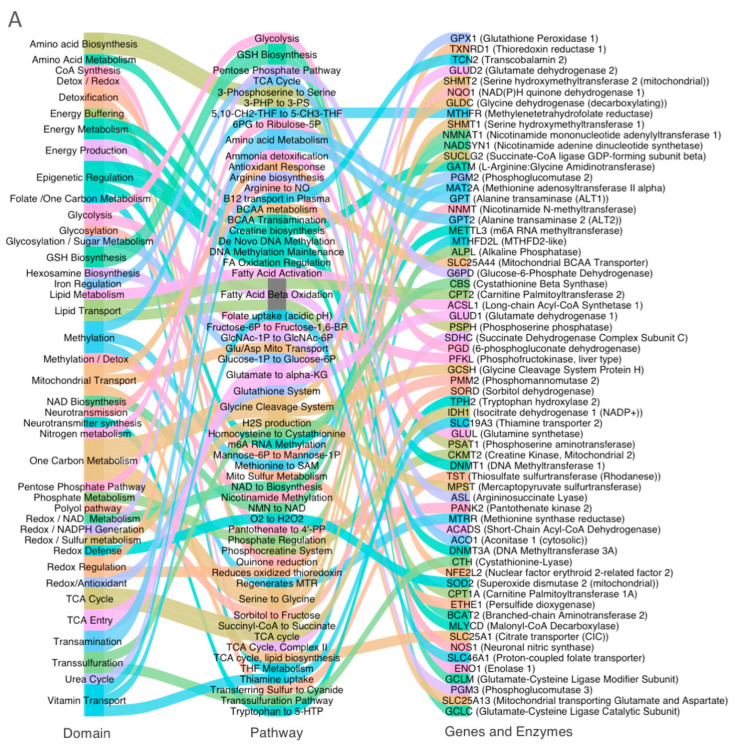
(**A**) Upregulated genes in ASD cortex. (**B**) Downregulated genes in ASD cortex.

**Figure 2 metabolites-15-00399-f002:**
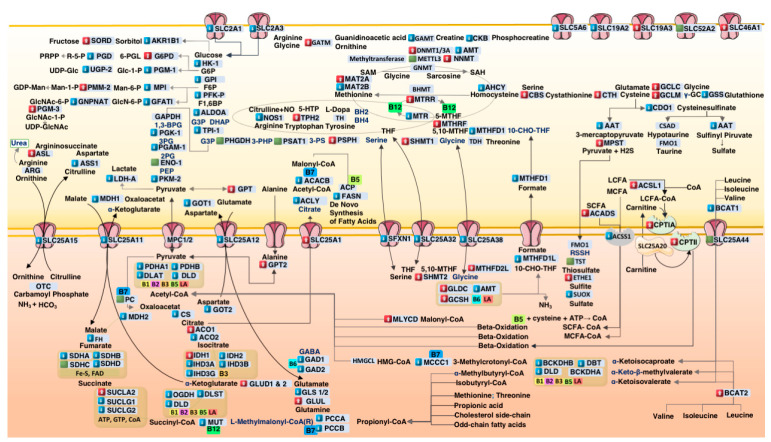
*Systems model of metabolic dysfunction in autism*. This integrative metabolic map illustrates core biochemical pathways affected in autism spectrum disorder (ASD), including mitochondrial energy production, amino acid catabolism, redox cycling, one-carbon methylation, and transsulfuration. Gene expression changes derived from ASD postmortem cortex samples are overlaid. Red boxes indicate increased mRNA transcription, blue boxes indicate decreased transcription, and green boxes represent minimal change (<1%). Transporters (e.g., SLC family) and cofactor-dependent enzymes (e.g., biotin [B7], thiamine [B1], lipoic acid [LA]) are localized to their respective subcellular compartments. Enzymes without transcriptional labeling indicate unavailable normalized TPM (nTPM) values in the reference dataset. Visual elements are adapted from Servier Medical Art (https://smart.servier.com, accessed on 11 May 2025), licensed under CC BY 4.0.

**Figure 3 metabolites-15-00399-f003:**
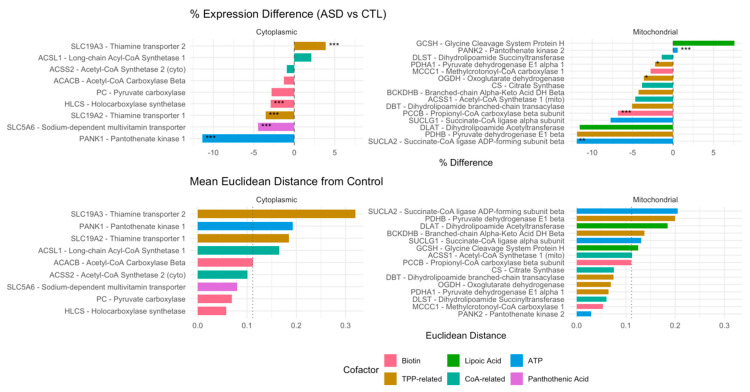
Cofactor-dependent transcriptomic vulnerability in autism. This figure highlights genes dependent on core mitochondrial and cytosolic cofactors: biotin, thiamine pyrophosphate (TPP), lipoic acid, pantothenic acid/CoA, and ATP. Top panels show the % expression difference (ASD vs. control) and bottom panels display the mean Euclidean distance from controls, reflecting multigene dispersion. Genes are stratified by compartment (cytoplasmic vs. mitochondrial) and are color-coded by cofactor dependency. Asterisks indicate statistical significance levels: *p* < 0.05 (*), *p* < 0.01 (**), and *p* < 0.001 (***).

**Figure 4 metabolites-15-00399-f004:**
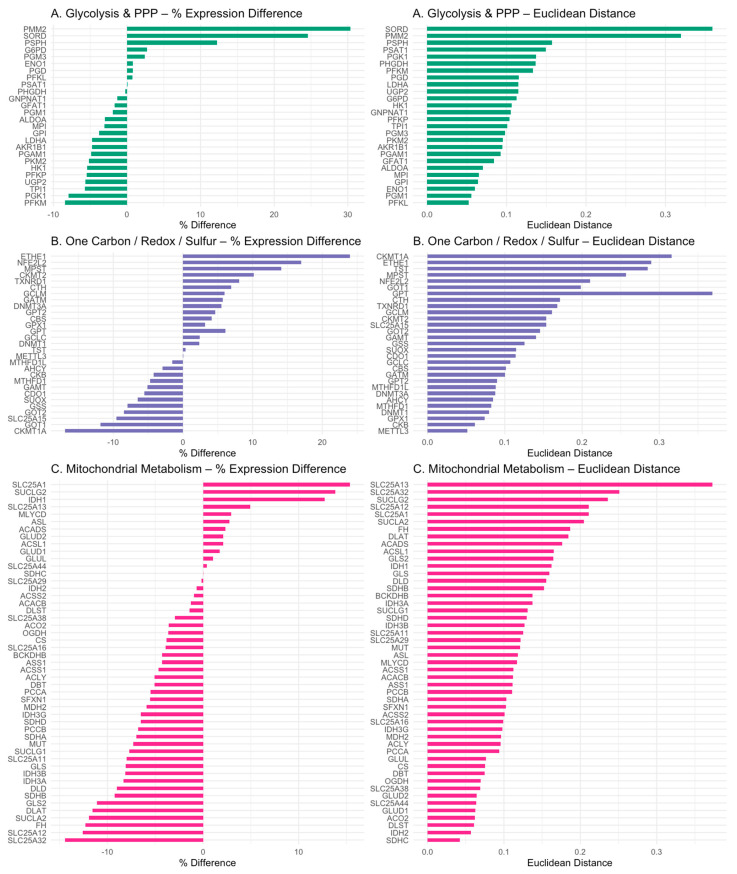
Domain-specific gene expression patterns. PM^3^ genes were grouped into three domains as follows: (**A**) glycolysis and pentose phosphate pathway (PPP), (**B**) one-carbon/redox/sulfur metabolism, and (**C**) mitochondrial metabolism. Percent expression differences (**left**) and mean Euclidean distances (**right**) are shown for each gene. Color-coding reflects domain identity. This visualization reveals selective transcriptional shifts and variability across functional pathways in the ASD cortex. Figure panels are generated in R using the ggplot2 and patchwork packages.

**Table 1 metabolites-15-00399-t001:** Differentially expressed metabolic genes in ASD cortex identified by the PM^3^ model. This table summarizes the top-ranked metabolic genes identified by the PM^3^ (Personalized Metabolic Margin Mapping) model as differentially expressed in the postmortem brain cortex of individuals with autism spectrum disorder (ASD) relative to neurotypical controls. Genes are ranked by their anomaly scores (AUC) derived from unsupervised Isolation Forest analysis, which reflect expression divergence between groups. Expression changes are presented as a percentage difference (%Δ), with positive values indicating upregulation and negative values indicating downregulation in ASD samples.

Gene Symbol	Gene Name	%Δ Expression	log2FC	AUC	Cofactor(s)	Pathway	Functional Role
SUCLA2	Succinate-CoA ligase (ADP-forming β)	−11.93	−0.180	0.88	ADP, CoA	TCA Cycle	Succinyl–CoA to Succinate
GOT1	Aspartate transaminase 1 (AST1)	−11.82	−0.181	0.86	PLP (Vitamin B6)	Transamination	Glutamate/Oxaloacetate Shuttle
SLC25A11	2-Oxoglutarate/malate carrier (OGC)	−7.99	−0.120	0.85		Mitochondrial Transport	Malate Shuttle (Mito)
CKMT1A	Mitochondrial creatine kinase 1A	−16.9	−0.267	0.84	Mg^2+,^ ATP	Energy Buffering	Phosphocreatine Shuttle
GPT2	Alanine transaminase 2 (ALT2)	4.64	0.065	0.83	PLP (Vitamin B6)	Transamination	Alanine ↔ Pyruvate
GOT2	Aspartate transaminase 2 (AST2)	−8.48	−0.128	0.80	PLP (Vitamin B6)	Transamination	Mitochondrial Transamination
GLS	Glutaminase	−8.08	−0.122	0.79	Glutamine	Glutamate Metabolism	Glutamine to Glutamate
IDH3A	Isocitrate dehydrogenase 3α(NAD^+^-linked)	−8.33	−0.125	0.79	NAD+	TCA Cycle	Isocitrate to α-Ketoglutarate
MDH2	Malate dehydrogenase 2 (mitochondrial)	−5.93	−0.088	0.78	NAD+	TCA Cycle	Malate to Oxaloacetate (mito)
IDH3B	Isocitrate dehydrogenase3β (NAD^+^-linked)	−8.13	−0.122	0.78	NAD+	TCA Cycle	Isocitrate to α-Ketoglutarate
SHMT1	Serine hydroxymethyltransferase 1	7.55	0.105	0.77	PLP (Vitamin B6)	One Carbon Metabolism	Serine to Glycine
MDH1	Malate dehydrogenase 1 (cytosolic)	−11.78	−0.181	0.74	NAD+	NAD Cycling/Redox	Malate to Oxaloacetate
OGDH	Oxoglutarate dehydrogenase	−3.63	−0.053	0.73	B1, NAD+, CoA, lipoate	Energy/TCA Cycle	α-Ketoglutarate to Succinyl-CoA
PDHA1	Pyruvate dehydrogenase E1 α subunit	−2.25	−0.033	0.73	B1, NAD+, CoA, lipoate	Energy/PDH complex	Pyruvate to Acetyl-CoA
MAT2B	Methionine adenosyltransferase II β	−11.9	−0.184	0.72		Methylation	Methionine to SAM (Regulation)
SFXN1	Serine transport into mitochondria	−5.51	−0.082	0.71		Mitochondrial Transport	Mitochondrial Serine Transport
MTHFD1	Methylene-THF dehydrogenase 1	−4.73	−0.070	0.71	NADH+	Folate/One-Carbon Metabolism	Folate Interconversion
DLD	Dihydrolipoamide dehydrogenase	−9.0	−0.136	0.71	FAD, NAD	Energy/Redox	E3 Subunit of PDH/OGDH Complexes
SUOX	Sulfite oxidase	−6.6	−0.097	0.70	Molybdenum, Cytochrome b5, and Heme	Sulfur Metabolism	Sulfite Oxidation
TST	Thiosulfate sulfurtransferase (rhodanese)	0.36	0.005	0.70		Detoxification	Thiosulfate Sulfurtransferase (Detoxification)
SUCLG2	Succinate-CoA ligase (GDP-forming β)	13.83	0.187	0.69	GDP, CoA	TCA Cycle	Succinyl-CoA to Succinate

## Data Availability

This study used publicly available de-identified transcriptomic data. Public RNA-seq data from Parikshak et al. (GSE64018) is accessible via NCBI GEO.

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
