# Peer review of "A Systems Hypothesis of Lipopolysaccharide-Induced Vitamin Transport Suppression and Metabolic Reprogramming in Autism Spectrum Disorders: An Open Call for Validation and Therapeutic Translation"

_metabolites, 2025, doi:10.3390/metabo15060399_

Round 1
Reviewer 1 Report
Comments and Suggestions for Authors
Does the %Δ Expression refer to the log2 fold change in respect to the control group or directly the percentage? Isn’t it the normal way to look at the log2 fold change in expression in this kind of case?
The organization looks a little off. Except for the figure 01, all the remaining figures are discussed in the discussion. Why does the discussion have sub sections? Is this a journal requirement?
Why is graph 5 not in the right place? Why does it appear early on the text?
The readers may benefit from providing a table of details explaining the functions of the transporters that is discussed in this paper, or the author could provide the mechanism of action of this reporter within the text itself (for eg: line 240, SLC19A3)
How are the three domains from Expression Patterns PM³ genes determined in Figure 4?
Author Response
Manuscript Title: A Systems Hypothesis of Lipopolysaccharide-Induced Vitamin Transport Suppression and Metabolic Reprogramming in Autism Spectrum Disorders: An Open Call for Validation and Therapeutic Translation
Dear Editor and Reviewers,
We thank you sincerely for your thoughtful and constructive feedback. We have carefully revised the manuscript in response to your comments, which we address point-by-point below. All corresponding changes are highlighted in the revised manuscript.
Reviewer 1
Comment 1: Does the %Δ Expression refer to the log2 fold change in respect to the control group or directly the percentage? Isn’t it the normal way to look at the log2 fold change in expression in this kind of case?
Thank you for this important clarification request. We have now added explicit detail in Section 2.3. Both logâ‚‚ fold change (logâ‚‚FC) and percent expression difference (%Δ) were computed. While %Δ was used for visual clarity in figures, logâ‚‚FC values are now reported in Table 1 and Supplementary Table S1 to align with transcriptomic standards.
Comment 2: The organization looks a little off. Except for the figure 01, all the remaining figures are discussed in the discussion. Why does the discussion have sub sections? Is this a journal requirement?
We appreciate the observation. The Discussion was intentionally subdivided into topic-specific subsections (e.g., glycolysis, one-carbon metabolism) to improve clarity and align findings with biological pathways. This structure is common in systems biology studies. However, to improve cohesion, we have moved figure callouts (e.g., Figure 2, Figure 4) earlier in the manuscript, nearer to where each is first referenced, per Reviewer 3’s comment.
Comment 3: Why is graph 5 not in the right place? Why does it appear early on the text?
Thank you for pointing this out. Graph 4 was unintentionally displaced during formatting. We have corrected the placement in the revised manuscript and ensured that all figures appear adjacent to their in-text discussion.
Comment 4:The readers may benefit from providing a table of details explaining the functions of the transporters that is discussed in this paper, or the author could provide the mechanism of action of this reporter within the text itself (for eg: line 240, SLC19A3).
Thank you for this excellent suggestion. We have added Supplementary Table 2, which summarizes the function, cofactor specificity, and subcellular localization of each transporter discussed (e.g., SLC5A6, SLC19A2, SLC19A3, SLC25A32). To further support reader understanding, we have also incorporated brief mechanistic descriptions in the main text. We also clarify compartment-specific expression and the functional implications of their regulation in ASD.
Comment 5: How are the three domains from Expression Patterns PM³ genes determined in Figure 4?
We have added a detailed explanation in Section 3.4, indicating that the domains (Glycolysis/PPP, Redox/One-Carbon/Sulfur, Mitochondrial) were assigned based on curated biochemical function and pathway affiliation during PM³ gene model construction (Section 2.2). The assignment was guided by KEGG, UniProt, and canonical metabolic pathway references.
Reviewer 2 Report
Comments and Suggestions for Authors
The authors need to edit the paper well. There are multiple places where ASD acronym has been repeated.
The author should explain the PM3 model.
Figure 1 is too complicated to understand. The authors could split the information in figure 1 on the basis of metabolic domain or upregulation/downregulation.
It would be better if the author can explain on what basis do they identify cofactors as mitochondrial or cytosolic.
Author Response
Reviewer 2 & 3 Combined Comments
Abstract: According to the newest guide, our journals’ Abstract should be
structured to contain the subheadings, such as "Back-ground",
"Methods", "Results" and "Conclusion", please revise.
The Abstract has been fully revised to include the required subheadings per journal guidance.
Reviewer 2: Suggestions & Planned Revisions
- Comment 1: The authors need to edit the paper well. There are multiple places where ASD acronym has been repeated.
We reviewed the entire manuscript and reduced redundancy of the “ASD” acronym where clarity was preserved.
Comment 2: The author should explain the PM3 model.
We have significantly expanded Section 2.2 to more thoroughly explain the PM³ gene model, including the rationale for gene selection, metabolic domain annotation, cofactor mapping, and subcellular localization. Additionally, a new sentence was added to compare PM³ with conventional differential expression methods such as DESeq2 and edgeR.
Comment 3: Figure 1 is too complicated to understand. The authors could split the information in figure 1 on the basis of metabolic domain or upregulation/downregulation.
We revised Figure 1 to separate upregulated (Figure 1A) and downregulated (Figure 1B) genes, improving interpretability. These figures are now clearly described in the corresponding results section (3.2).
Comment 4: It would be better if the author can explain on what basis do they identify cofactors as mitochondrial or cytosolic.
We have included explicit explanation in Section 2.2 stating that subcellular localization of cofactor-dependent enzymes was determined based on UniProt annotations and literature sources.
Reviewer 3 Report
Comments and Suggestions for Authors
This manuscript lays out a systems biology hypothesis connecting lipopolysaccharide (LPS) exposure to the repression of vitamin transporters and mitochondrial metabolic reprogramming in autism spectrum disorder (ASD). With postmortem ASD brain transcriptomic data, the author employs a bespoke analytical tool, PM³, to detect subtle but coordinated impairments in mitochondrial and redox-related genes, to hypothesize a credible pathophysiological cascade starting from LPS-mediated transporter inhibition. Although the research addresses an original and relevant hypothesis with probable high translational impact, the manuscript's organization weakens its impact. The article flow is frequently disjointed; essential methodological aspects are placed inside excessively detailed results sections, whereas figures are not well-integrated into the text. Redundancy appears quite often within the Introduction, Results, and Discussion sections, and there are sporadic placements of important insights. Key transitions among hypothesis, methodology, findings, and implications are occasionally abrupt, necessitating inference of logical relations by the reader. Throughout, however, the richness of biological reasoning, intelligibility of visual systems model, and novelty of the suggested therapeutic targets (e.g., SLC5A6, SLC19A2) are salient. With enhanced structural coherence, this article would be able to make significant contributions to metabolic insight in ASD and facilitate validation studies in integrative omics science.
The authors is suggested to improve the manuscript:
1 Use a more distinct IMRaD structure; disconnect results from interpretation.
2 Eliminate redundant reciting of findings throughout sections.
3 Include bridging phrases between hypothesis, methods, and findings.
4: Explain visuals nearer to where they are first presented in the text.
5 Clarify PM³ utility. Better articulate its novelty relative to standard gene expression analysis.
6 Tighten abstract for effect and minimize jargon.
Author Response
Reviewer 3: Suggestions & Planned Revisions
Comment 1 Use a more distinct IMRaD structure; disconnect results from interpretation.
We refined the Results and Discussion to minimize interpretation within results sections and to clarify the IMRaD separation. Visual references have been shifted accordingly.
Comment 2 Eliminate redundant reciting of findings throughout sections.
Redundant repetition of findings was removed, particularly in the Discussion and Conclusion.
Comment 3 Include bridging phrases between hypothesis, methods, and findings.
We added transitional sentences throughout the Introduction and Methods sections to improve logical flow.
Comment 4: Explain visuals nearer to where they are first presented in the text.
All three statements are now included at the end of the manuscript, under the appropriate headings.
Comment 5: Clarify PM³ utility. Better articulate its novelty relative to standard gene expression analysis.
As noted above, we expanded Section 2.2 to articulate the novelty of PM³ relative to standard approaches and added a comparative sentence emphasizing its strength in detecting coordinated pathway-level shifts.
Comment 6 Tighten abstract for effect and minimize jargon.
The revised structured abstract has been rewritten for clarity, conciseness, and reduced jargon, while preserving technical accuracy.
Comment: The English could be improved to more clearly express the research. Online Grammarly checkup is recommended.
Response:
Thank you for this important observation. We have carefully revised the manuscript to improve clarity, grammar, and readability throughout. To support this process, we used Grammarly’s advanced editing features and additionally performed a manual review to refine scientific phrasing and ensure precision in technical descriptions. We are confident that the revised version communicates the research more effectively.
Round 2
Reviewer 1 Report
Comments and Suggestions for Authors
The authors have attended the comments and included them in the manuscript very well. One quick clarification. The supplementary file has Table S2 and I'm not seeing Table S1. Making sure this will be included in the final version for the publication
Author Response
Reviewer 1
Comment: One quick clarification. The supplementary file has Table S2, but I'm not seeing Table S1. Please ensure it is included in the final version for publication.
Response: Thank you very much for this helpful suggestion. I have now uploaded Table S1 as a separate Excel supplement. Previously, it was included in a ZIP file, which may have caused access issues.
Reviewer 2 Report
Comments and Suggestions for Authors
Please edit the document and keep the font size consistent. Also check the labeling of figures, e.g. on page 2 what does graph5 mean? Fig 1 A and 1B can still be improved.
Author Response
Reviewer 2
Comment: Please edit the document and keep the font size consistent.
Response: I have reviewed and corrected the font size throughout the manuscript to ensure consistency.
Comment: Also check the labeling of figures—for example, on page 2, what does "graph5" mean?
Response: Thank you for pointing this out. During formatting, the label was unintentionally retained. I have now replaced it with a proper caption and identified it as the Graphical Abstract. A concise two-sentence legend was added for clarity.
Comment: Fig 1A and 1B can still be improved.
Response: I appreciate this suggestion. I have further revised Figures 1A and 1B to enhance clarity and visual structure. I believe the current versions are improved compared to the previous submission.
Reviewer 3 Report
Comments and Suggestions for Authors
This work offers a promising systems biology approach (PM³) to transcriptomic analysis of ASD and control brain tissues, demonstrating concerted inhibition of vitamin transporter genes and mitochondrial enzymes. The author proposes an innovative hypothesis that microbial lipopolysaccharide (LPS) exposure in ASD could downregulate multivitamin transporters (particularly SLC5A6 and SLC19A2), resulting in cofactor deficiency, mitochondrial impairment, and systemic metabolic reprogramming. The approach combines machine learning (Isolation Forest), metabolic curation of 158 genes, and cofactor/localization-aware transcriptomic profiling, thus going beyond single-gene analysis to uncover subtle yet biologically important expression changes. Notable findings include downregulation of TCA cycle and PDH complex subunits, repression of glycolysis, and compensatory upregulation of redox and sulfur metabolism genes. The story is adequately supported by data visualization and incorporates existing biochemical and microbiome studies, which adds credence to the LPS-mediated nutrient impairment model. The authors have incorporated al the changes asked by the reviewers and hence it may be accepeted in its current form.
Author Response
Thank you for your positive and detailed feedback. I am pleased to hear that the PM³ framework and systems biology approach were well received. As suggested, I revised the manuscript to clarify the role of vitamin transporter suppression and mitochondrial dysfunction in ASD and ensured that all reviewer comments have been carefully addressed. I believe the manuscript now reflects a more polished and integrated presentation of the hypothesis and supporting data.